# Red Beetroot and Betalains as Cancer Chemopreventative Agents

**DOI:** 10.3390/molecules24081602

**Published:** 2019-04-23

**Authors:** John F. Lechner, Gary D. Stoner

**Affiliations:** Department of Microbiology and Immunology, Montana State University, Bozeman, MT 59717, USA; gstoner@montana.edu

**Keywords:** beetroot, betanins, cancer chemoprevention, anti-oxidant, inflammation, apoptosis

## Abstract

Carcinogenesis is the process whereby a normal cell is transformed into a neoplastic cell. This action involves several steps starting with initiation and followed by promotion and progression. Driving these stages are oxidative stress and inflammation, which in turn encompasses a myriad of aberrant gene expressions, both within the transforming cell population and the cells within the surrounding lesion. Chemoprevention of cancer with bioreactive foods or their extracted/purified components occurs via normalizing these inappropriate gene activities. Various foods/agents have been shown to affect different gene expressions. In this review, we discuss whereby the chemoprevention activities of the red beetroot itself may disrupt carcinogenesis and the activities of the water-soluble betalains extracted from the plant.

## 1. Introduction

Carcinogenesis is the series of events whereby a normal cell is transformed into a neoplastic form. Driving these stages in vivo is continued oxidative stress, which causes a myriad of more than 500 aberrant gene expressions both in the target cells and the lesion’s microenvironment [1,2]. Continued oxidative stress is the imbalance between oxidants, e.g., reactive oxygen species (ROS), and their elimination by the cell’s protective mechanisms. Endogenous ROS derives primarily from the NADPH oxidases, the mitochondria, and peroxisome oxidation of proteins [2,3,4,5]. ROS in proper amounts are regulators of homeostatic signal transduction [6,7,8]. Continued excess ROS produced under sustained oxidative stress caused by biological, chemical, or physical factors results in inflammation [2,9,10,11,12]. Inflammation arises as a series of pathologic events, including the invasion of neutrophils, lymphocytes, and macrophages, which are sources of pro-inflammatory cytokines and ROS, and the influx of fibroblasts into the lesions [9]. This sets up a chronic active situation that promotes genomic instability and mutation, altered epigenetic events, inappropriate gene expression, changed microRNA translation, enhanced proliferation of the initiated cells, enabling resistance to cell death, antagonism of immune surveillance, and neo-vascularization [13]. All of these are targets for chemopreventive agents, and numerous reports of various bioactive foods and their extracted compounds have been shown to affect these hallmarks of carcinogenesis. Interestingly, a common feature of these foods and purified compounds is that they exhibit anti-oxidant and anti-inflammatory activity [14,15]. The chemopreventive properties of beetroot and its constituents is no exception.

## 2. Bioactive Compounds in Beetroot

Red beetroot is a vegetable rich in carbohydrates, fat, micro-nutrients, and constituents with bioactive properties [16]. The bioactive constituents include betaine, polyphenols, carotenoids, flavonoids, saponins, and the water-soluble pigments the betalains [17] (see Table 1).

### 2.1. Betaine

Betaine is a vital methyl group donor [18]. Betaine can disrupt inflammation [18,19,20] through suppressing nuclear factor kappa-light chain enhancer of activated B cells (NF-κB) and Akt activation [51], and the initiation of inflammasomes [18]. Betaine has not been directly investigated as a cancer chemoprevention compound, although its anti-inflammation effects suggest that the compound may be active. Betaine is readily taken up by ingestion in humans [18].

### 2.2. Polyphenols

Polyphenols are a structural class of chemicals characterized by the presence of large multiples of phenol structural units. The number and characteristics of these phenol structures underlie the unique physical, chemical, and biological properties. The polyphenols in red beetroot have chemopreventive qualities, as they have hydroxyl groups that donate their protons to ROS. The phenolic acids, e.g., ferulic acid, caffeic acid, *p*-coumaric acid, syringic acid, and vanillic acid have been purified from beetroot [17]. While none of these compounds have been extracted from red beetroot and evaluated for chemopreventive potential, data derived about these compounds extracted from other sources (see below) suggest that the polyphenols in red beetroot should have chemopreventive qualities.

#### 2.2.1. Phenolic Acids

Ferulic acid (FA) is a strong membrane antioxidant and known to be an effective scavenger of free radicals [21]. It has been shown to induce cell apoptosis through loss of mitochondrial membrane potential [22]. Twenty-five FA metabolites were found in the plasma, most of which were also found in the urine, while in the feces, colonic metabolism led to simpler phenolic compounds [21]. Ferulic acid intake reduced NADPH oxidase activity, superoxide release, apoptosis, and necrosis in peripheral blood mononuclear cells in mice [22]. Superoxide production was abrogated by FA in rats [23]. This polyphenol was also shown to improve lung inflammation in rats [52,53] suggesting that it may antagonize lung carcinogenesis.

Caffeic acid absolute bioavailability in rats was 14.7%, and its intestinal absorption was 12.4% [24]. Prasad et al. [25] reported that caffeic acid significantly reduced lipid peroxidation and decreased DNA damage in UVB-irradiated lymphocytes. Moreover, caffeic acid was shown to reduce oxidative stress and inflammation induced by 12-*O*-tetradecanoyl-phorbol-13-acetate (TPA) in vivo in mouse skin. The compound significantly inhibited TPA induced lipid peroxidation and tumor necrosis factor alpha (TNF-α) release, up-regulated glutathione content, and inhibited TPA induced expression of NF-κB and COX-2 [26].

*P*-coumaric acid (*p*-CA) is easily and quickly absorbed in the upper gastrointestinal tract and is excreted in the urine [54]. In vitro studies have indicated that *p*-CA moderately inhibits the growth of some tumor cell lines. *p*-CA inhibits the AKT and ERK signaling pathways that are responsible for angiogenesis and reduces the mRNA expression of VEGF and bFGF, two of the most important angiogenic factors that stimulate the permeability, proliferation, and migration. In a mouse xenotransplant model intraperitoneal administration of *p*-CA for one week significantly decreased tumor volumes by reducing angiogenesis within the tumor [27]. *p*-CA also exerts moderate protection against 1,2-dimethylhydrazine-induced colon cancer in rats, and has been shown to impede aberrant crypt development in 1,2-dimethylhydrazine (DMH)-treated animals [28]. The compound was found to down-regulate the cell cycle regulating genes cyclin B1, cdc2 and mdm3 and the immediate early response genes c-fos, c-jun, and c-myc in *p*-CA and DMH treated rats verses rats that only received DMH. The observed apoptosis correlated with up-regulation of Bax. *p*-CA also augmented detoxification in the colon by modulating the nuclear to cytoplasmic ratio of Nfr2 and the accompanying up-regulation of the down-stream phase II xenobiotic detoxifying genes heme oxygenase (decycling) 1, Uridine 5′-diphospho-glucuronosyltransferase, NAD(P)H dehydrogenase [quinone] 1, thioredoxin, and glutathione *S*-transferase [29].

Syringic acid bioavailability was found to be 86.27% in the blood of rabbits. It has been shown to relatively non-toxic to animals [55]. It was found to have potent chemoprevention activity using a UV-induced skin carcinogenesis mouse model. The compound was also studied in a pre-cancerous human skin cell line model. The study showed that UV-induced activation of COX2, matrix metalloproteinase-a, prostaglandin E_2_, and activator protein-1 activity was inhibited by syringic acid. Moreover, the induction of protein tyrosine phosphatase-b1, which activates EGFR, was inhibited. This phenol also knocked down the mitogen-activated protein kinases and Akt pathways. It also inhibited UV induction of NADPH oxidase activity [56]. Vanillic acid has been studied in a rat model of endometrial carcinoma [30]. The animals treated with vanillic acid showed normalization of the histopathological lesions. In addition, levels of cytochrome P450 were reduced, while phase II enzyme activities were increased. Additionally, a decrease in matrix metalloproteinases 2 and 9, as well as the cell cycle regulator cyclin D was found.

#### 2.2.2. Flavonoids

Flavonoids have the general structure of a 15-carbon skeleton, which consists of two phenyl rings and a heterocyclic ring. The primary flavonoids in red beetroot are rutin, kaempferol, rhamnetin, rhamnocitrin, and astragalin. The relatively poor bioavailability of rutin has been discussed by Faggian et al. [40]. Rutin was shown to decrease focal areas of dysplasia in mice chemically-induced to develop colon cancer [31]. It was also shown to decrease oxidative stress and inflammation in a rat liver model. In addition, rutin-treated animals showed down-regulation of the inflammatory markers TNF-α, IL-6, p38-MAPK, NF-κB, i-NOS, and COX2 [32]. The relatively poor bioavailability of kaempferol has resulted in the chemopreventive activity being shown primarily in in vitro studies with cancer cell lines. For example, it inhibited the function of phosphorylated AKT, CyclinD1, CDK4, Bid, Mcl-1 and Bcl-xL, and promoted the expression of *p*-BRCA1, *p*-ATM, p53, p21, p38, Bax and Bid, while triggering apoptosis and S phase arrest in a study of bladder cancer cells [33]. The bioavailability of rhamnetin has not been studied. The compound has been shown to be an anti-oxidant and anti-inflammatory [34]. Rhamnocitrin bioavailability has also not been reported. It exhibited significant cytotoxicity against HeLa cells [35]. Astragalin was shown to be cytotoxic to cultured human hepatocellular carcinoma HepG2 cells, but not human breast cancer cell line MCF-7 [57]. It also decreased cell viability and increased apoptosis of cultured human lung cancer cell lines (A549 and H1299), but not the normal (BEAS-2B) lung cells [58]. The mechanism for its selectivity toward tumor versus normal cells is unknown, but anti-oxidants can inhibit the growth of tumor cells through cell cycle arrest or apoptosis via depletion of reactive oxygen species because their growth is dependent on H_2_O_2_ [14]. NF-κB activity was also antagonized by rhamnocitrin in TNF-α treated A549 lung tumor cells. Rhamnocitrin inhibited growth of these cells when xenotransplanted into athymic nude mice. The tumors that developed in the rhamnocitrin-treated mice showed significant necrosis and apoptosis [36].

#### 2.2.3. The Triterpene Saponins

Triterpenoid saponins are triterpenes containing 30 carbon atoms. Some triterpenes are steroidal in nature. These sugars can be cleaved off in the gut by bacteria, sometimes allowing the aglycone to be absorbed into the bloodstream and inserted into cell membranes. The saponins in red beetroot are oleanolic acid and several betavulgarosides. Koczurliewicz et al. [37] published an overview of the anti-tumor effects of triterpene saponins and the biological activities of oleanolic acid have been reviewed by Ayeleso et al. [38]. Recently, the molecular actions of oleanolic acid on tumor suppressive activity have been shown for several xenotransplanted cell lines. Cell cycle analysis revealed that oleanolic acid induced cell cycle arrest in HepG2 cells at the sub-G1 (apoptotic) phase of the cell cycle, in a dose-dependent manner [39]. Kim et al. [59] showed that the compound increased apoptosis and decreased cell cycling in xenotransplanted prostate DU145 cells. Activation of the genes p53, Bax, and Akt was observed, while cyclin B1, cyclin E, cdk2, *p*-erk, and c-jun activities were down-regulated.

## 3. Carotenoids

Carotenoids, also called tetraterpenoids, are organic pigments. The carotenoids in red beetroot are β-carotene and lutein. The anti-cancer properties of these compounds have been recently reviewed [60]. The vast majority of the studies with β-carotene have been with tumor cell lines. While these investigations are informative as to the elucidation of the molecular mechanisms of these compounds, it is somewhat difficult to extrapolate the results to chemoprevention processes, because both the premalignant target cell and the cells in the microenvironment are involved the tumorigenesis. However, these studies suggest that one mechanism of chemoprevention in vivo is the selective killing of tumor cells. The interesting question is: Why these agents do not cause normal cells to initiate apoptosis, as well? β-carotene acts as an anti-oxidant. It also inhibits cell proliferation, arrests the cell cycle, and increases apoptosis, as well as decreasing the percentage of Bcl2-positive cells, while increasing the level of p53 [40]. Lutein is a strong anti-oxidant. It was shown to increase IL-6 expression in activated macrophages, and to up-regulate expression of COX-2 in premalignant human keratinocytes. It enhanced IL-2 and IFN-γ production in mice [40]. Dietary lutein decreased mammary tumor growth and Bcl2 expression, while up-regulating pro-apoptotic genes p53 and BAX in the BALB/c mouse [60].

## 4. Non-Defined Red Beetroot Components with Chemopreventive Activity

Fibrous material from red beetroot, prepared by extraction of the dyes and water-soluble components, was found to significantly reduce the incidence of precancerous liver lesions and the number of animals bearing tumors in a rat model [41]. In addition, pectin extract from sugar beets was shown to be cytotoxic to MCF-7 cells [61]. These two observations suggest that red beetroots have anti-cancer compounds that are yet to be purified and identified.

## 5. Betanins

The most studied bioactive compounds in beetroot are the betalains. Betalains are a class of red and yellow indole-derived pigments found in plants of the *Caryophyllales*, where they replace anthocyanin pigments. The predominant forms of these water-soluble pigments are betacyanin (red color) and betaxanthin (yellow color) [62]. Red beetroot is the primary source of betalains in western diets, as they are not widely present in the plant world [63]. They are not carcinogenic [64] or mutagenic [42], and are anti-mutagenic against the direct acting mutagen, Methylnitro-nitrosoguanidine (MNNG) as assessed by the Ames test [42,65]. They did not induce demethylation in the promoters of tested methylation-silenced onco-suppressor genes in MCF-7 cells [66]. Betalains have no known toxicity in rodents and humans [62,63], and the extracted food dye E162 (which is primarily betalains) is commonly used in the food industry [62,63].

### 5.1. Betanins Bioavailability

Consumed betalains in humans were found to be maximal in plasma in three hours and the compound was undetectable by eight hrs. Excretion of betanin occurs rapidly in rats, with a half-life of 32 min in plasma and the urine was colored in three min. Stomach wall, small intestine and colon metabolized 75%, 35%, and 60% of the added betanin, respectively. In general, betalains have poor bioavailability, and renal clearance is a minor route of their elimination. The agents are primarily metabolized and degraded in the gastrointestinal cells of humans. Finally, it has been shown that betanins are absorbed by intestinal epithelial cells through paracellular junctions, whereby the compound enters the blood stream and incorporates into red blood cells and lipoproteins [43,62,63,67,68].

### 5.2. Anti-Oxidant and Anti-Inflammatory Activity

The radical scavenging activity of betalains has been recently reviewed by Rahimi et al. [49] and Ninfali et al. [45]. Betalains have strong free radical scavenging, and anti-oxidant activities. Their scavenging activity is comparable to butylated hydroxytoluene, the widely used synthetic anti-oxidant [69]. Betalains scavenge galvinoxyl, hydroxyl, and superoxide free radicals [69]. In vitro DNA damage in cultured human liver hepatoma cells caused by exogenous H_2_O_2_ was reduced in cells incubated with betalains [63]. The compound also significantly inhibited ROS production by cultured human, neutrophils, and decreased DNA damage in the activated cells [68,70,71,72]. Betalains have anti-inflammatory activity, as well. Intraperitoneal treatment with betanins diminished carrageenan-induced paw edema and neutrophil migration to the paw skin tissue. Oral betalains also inhibited recruitment of total leukocytes, including mononuclear cells and neutrophils. Furthermore, betalain significantly reduced carrageenan-induced superoxide anion, tumor necrosis factor-alpha (TNF-α) and interleukin (IL)-1β levels in the peritoneal fluid, as well as augmenting IL-10 levels [73]. Allegra et al. [74] reported that betanins effectively scavenged hypochlorous acid (HClO), the most powerful oxidant produced by neutrophils. HClO has been demonstrated to be a local mediator of tissue damage and the resulting inflammation in various inflammatory diseases [75,76]. Red beetroot dye extract also attenuated renal dysfunction through the reduction of oxidative stress, inflammation, and apoptosis in the kidneys of rats [77].

### 5.3. Phase II Detoxifying Enzyme Activity

Betalains were shown to induce the phase II detoxifying enzyme quinone reductase in murine hepatoma cells in vitro [47]. In rats, long-term feeding (28 days) with red beetroot juice on liver injury induced by the hepatocarcinogen N-nitrosodiethylamine (NDEA) was studied. Combined treatment with betalains and the carcinogen significantly enhanced the activity of NAD(P)H dehydrogenase [quinone] 1 (NQO1). Red beetroot juice also reduced the DNA damage caused by NDEA treatment, as well as other biomarkers of liver injury [78]. These results were followed with a study whereby betanin was found to induce Nrf2 and down-stream phase II detoxification genes in liver cell lines. The influence of betanins on the activation of Nrf2 and the expression of GSTA, GSTP, GSTM, GSTT, NQO1, and HO-1 was assessed in human non-tumor THLE-2 and hepatoma-derived HepG2 hepatic cell lines. Treatment of both cell lines resulted in the translocation of Nrf2 from the cytosol to the nucleus. In the THLE-2 cells this was associated with the phosphorylation of Akt, c-JNK, and ERK proteins, and a significant increase in the mRNA and protein levels of GSTP, GSTT, GSTM, and NQO1. Conversely, other than the translocation of Nrf2 from the cytosol to the nucleus, betanin did not modulate any of these parameters in the HepG2 cells [78]. These results indicate that betanin activation of Nrf2 and subsequent induction of the expression of genes controlled by this factor may exert hepatoprotective and anticarcinogenic effects in normal tissues.

### 5.4. Betanins Cytotoxicity of Cultured Cells

We have found that low doses of betanins (<100 μg/mL) are not cytotoxic to SV40 t-antigen- and papilloma virus-immortalized human lung epithelial cells. On the other hand, betanins are cytotoxic at low concentrations and in a dose-response fashion to some cultured tumor cells, but not all. Nowacki et al. [71] suggested that betanin-induced apoptosis was dependent on p53 status, i.e., normal p53 imparts resistance to betanin cytotoxicity. Kapadia et al. [43] reported that growth of several human cell lines was not inhibited by low doses of the commercial dye form of betanins, E162. With regard to apoptosis, betanin did not affect the activity of caspase-3 in resting neutrophils, but significantly enhanced the enzyme’s activity in stimulated cells [72].

Treatment of colorectal adenocarcinoma (Caco-2) cells with H_2_O_2_ significantly increased DNA strand breaks. Strand break rejoining is normally rapid, but treatment with H_2_O_2_ induces additional breaks and slows down the repair process. Pretreatment with betanin accelerated this repair process and reduced DNA damage in these cells [46]. A study to evaluate the effect of betanin on ROS production, DNA damage and apoptosis in resting and phorbol 12-myristate13-acetate (TPA)-stimulated human polymorphonuclear neutrophils showed that betanins significantly inhibit ROS production. Betanin also decreased the percentage of DNA breaks in stimulated neutrophils, whereas resting neutrophils exhibited an increased level of DNA breaks [46].

Betanins have also been evaluated for their ability to suppress lipid accumulation in 3T3-L1 adipocytes. Betanin, in the range from 10–100 µM did not elicit any cytotoxicity effect on cell survival. However, betanin significantly inhibited adipocyte cell differentiation and lipid accumulation [79]. Since obesity increases cancer risk, one possible mechanism for betanins chemoprevention could be related to impeding this risk factor.

### 5.5. Chemoprevention with Betanins

There have not been any published human cancer chemoprevention studies with red beetroot or the bioactive molecules extracted from the plant. However, as noted above, Bobek et al. [41] showed that extracted beet root fiber reduced pre-cancerous liver lesions in rats, suggesting that the non-aqueous bioreactive compounds may be chemopreventive. Regarding betalains, extracted preparations were showed to inhibit xenotransplanted tumor growth in athymic nude mice when injected intra peritoneally [48,71]. In addition, Kapidia et al. [43,80,81] found that drinking water supplemented with a low concentration of red beetroot dye extract (food color dye E162) significantly inhibited the induction of lung, skin, and liver tumors in carcinogen-treated mice. We [50] showed that E162-drinking water inhibited esophageal carcinogenesis in carcinogen-treated rats with a concomitant reduction of inflammation and angiogenesis, and increased apoptosis. Zhang et al. [82] reported that betanins in the drinking water antagonized lung carcinogenesis in carcinogen-treated mice, with the reduction in angiogenesis and increased rate of apoptosis via activated caspase-3, -7, and -9 and PARP.

## 6. Conclusions and Future Reaearch

It is clear that betanin pigments (and possibly other soluble constituents, e.g., betaine in dye E162) are effective at cancer chemoprevention at very low doses in drinking water consumed by animals *ad libitum*. The wealth of bioreactive agents within red beetroot itself [16,17,83] (see Table 1) argues that chemoprevention studies using a diet supplemented with beet powder or root chips could well be chemopreventive in carcinogen-treated animals. In fact, Guldiken et al., [84] found that processed red beets showed significant anti-oxidant capacity. Thus, this supposition should be tested. If these studies are positive, human chemoprevention trials would be warranted, especially if the beet formulations are found to have a favorable toxicity profile. In addition, as tumors developed in the betalain-treated animals [43,50,82], it would be interesting to ascertain if the combination of red beetroot powder or chips coupled with betanins-containing (E162 dye) drinking water would be more effective in reducing tumor number and tumor burden than the powder of chips alone. This combination study would answer the question as to whether betalains or phenolic compounds have different but complimentary chemopreventive mechanisms. The occurrence of betanins in nature is mutually exclusive from the anthocyanins [16,17], and betanins and anthocyanins which are chemopreventive [85]), are structurally and chemically unlike, as betalains contain nitrogen whereas anthocyanins do not. Thus, animal studies with a diet containing both anthocyanins and betalains should be undertaken. Human chemoprevention trials would also be warranted if these combination studies are positive. Alternatively, because the chemicals in the red beetroot plant, the anthocyanins, and the water soluble betanin pigments are all anti-oxidants, they may all ultimately constrain carcinogenesis via the same ROS-molecular mechanisms. If so, red beetroot powder, anthocyanins, and E162 dye would be equally effective in reducing carcinogenesis and their combination would not inhibit the development of tumors any better that the individual agents. Alternatively, the different agents may have different pharmacokinetics and the combination of agents could be additive or synergistic. This possibility of enhanced chemopreventive activity of combined agents should be tested.

The molecular mechanism(s) of betanins chemoprevention (see Figure 1) has not been elucidated. However, inflammation due to ROS and HClO released from neutrophils is neutralized by betanins [74], thereby reducing oxidative stress. How betanins inhibit ROS production by neutrophils [46] has not been elucidated. Betanins also reduce the number of infiltrating neutrophils [44,73] into inflammatory lesions by an unknown mechanism, and new blood vessel formation (angiogenesis) in tumors [50]. These observations suggest that one of the inhibitory mechanisms of betanins is to negate the development of stromal elements in inflammatory lesions. Betanins are non-mutagenic [65] and affect the molecular mechanisms of transformed cells, primarily by reducing their growth rate and inducing apoptosis [50,82]. Thus, it can be concluded that betanins affect both the tumor environment and the tumor cell itself. Understanding how this is occurring is the focus of future investigations. Additionally, experiments ascertaining why some tumor cells escape the inhibitory effects of betanins should be informative.

## Figures and Tables

**Figure 1 molecules-24-01602-f001:**
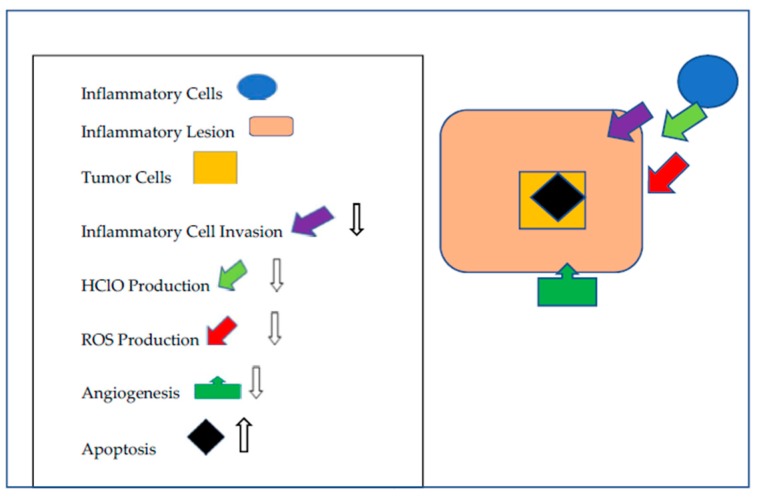
Mechanisms by which that Betanins Augment Cancer Chemoprevention. Betanins impede inflammatory lesion development by limiting inflammatory cell invasion, the release of HClO and the neutralization of ROS produced by the inflammatory cells and the tumor cells. Angiogenesis is moderated and apoptosis is increased.

**Table 1 molecules-24-01602-t001:** Compilation of Bio-reactive Compounds in Red Beetroot with Cancer Chemopreventive Activity.

Compound	Beneficial Effects	References
Betaine	Disrupt inflammation	[18,19,20]
Ferulic Acid	Membrane antioxidant, reduce NADPH oxidase activity and superoxide production	[21,22,23]
Caffeic Acid	Reduced lipid peroxidation, oxidative stress, inflammation	[24,25,26]
*p*-Coumaric Acid	Inhibits AKT and ERK signaling, angiogenesis, up-regulation of phase II enzymes, apoptosis	[27,28,29]
Syringic Acid	Chemoprevention activity in mice, Inhibit EGFR, Akt and NADPH oxidase activities	[30]
Rutin	Chemoprevention activity in mice, down-regulation of inflammation	[31]
Kaempferol	Apoptosis in vitro	[32,33]
Rhamnetin	Anti-oxidant, anti-inflammatory	[34]
Rhamnocitrin	Cytotoxicity	[35]
Astragalin	Inhibited growth of xenotransplanted tumor cells in nude mice	[36]
Oleanolic Acid	Inhibited growth of xenotransplanted tumor cells in nude mice	[37,38,39]
β-carotene	Anti-oxidant, increased apoptosis	[40]
Lutein	Anti-oxidant, activated macrophages, up-regulated pro-apoptotic genes	[40]
Non-defined	Reduced pre-cancerous lesions in rat	[41]
Betalains	Anti-oxidant, decreased inflammation, increased apoptosis, anti-mutagenic, induce phase II enzymes, chemopreventive activity	[42,43,44,45,46,47,48,49,50]

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
