# Peer review of "Red Beetroot and Betalains as Cancer Chemopreventative Agents"

_molecules, 2019, doi:10.3390/molecules24081602_

Reviewer 1 Report

Comments to the Authors:

The manuscript entitled “Red beetroot and betalains as cancer chemoprevetative agents” have demonstrated significant review from characterization of bioactive compounds of beetroot. The study present significant results in biological properties of phytochemicals fields. This study is a summary of current information on beetroot phytochemicals and their anticancer properties and present significant results in biological properties of phytochemicals fields. Moreover, on the basis of available information, Authors try to partially explain the mechanism of anticancer activity of beetroots phytochemicals.

Authors should correct manuscript according to the suggestion and completed some information.

Minor issues:

Chapter: Bioactive compounds and beetroot

Line 68 and 91: As in other phytochemicals cases names abbreviations should be defined

Line 151: Authors should characterized antioxidant activity of polyphenols isolated from beetroot as in case of betanins. Example publication:

Guldiken et al., 2016, Home-processed red beetroot (Beta vulgaris L.) products: changes in antioxidant properties and bioaccessibility, Int. J. Mol. Sci. 17, 858

Line 152 the numeration of chapter should be changes, In my opinion “Non-phenol red beetroot components with chemopreventive activity” should be another, separate chapter 2.4, not “Polyphenols” part

More important information on the anticancer mechanism of individual groups of phytochemicals should be collected in the tables with appropriate reference.

References:

Line 293: title of journal in reference no. 3 should be corrected

Line 398: title of journal should be italic

Author Response

The manuscript entitled “Red beetroot and betalains as cancer chemoprevetative agents” have demonstrated significant review from characterization of bioactive compounds of beetroot. The study present significant results in biological properties of phytochemicals fields. This study is a summary of current information on beetroot phytochemicals and their anticancer properties and present significant results in biological properties of phytochemicals fields. Moreover, on the basis of available information, Authors try to partially explain the mechanism of anticancer activity of beetroots phytochemicals.

Authors should correct manuscript according to the suggestion and completed some information.  Done so in red font.

Minor issues:

Chapter: Bioactive compounds and beetroot

Line 68 and 91: As in other phytochemicals cases names abbreviations should be defined

Response: As the molecule is only written one time, the abbreviation is not necessary. Wher we have written the name more than one time, we have included the abbreviation.

Line 151: Authors should characterized antioxidant activity of polyphenols isolated from beetroot as in case of betanins. Example publication:

Guldiken et al., 2016, Home-processed red beetroot (Beta vulgaris L.) products: changes in antioxidant properties and bioaccessibility, Int. J. Mol. Sci. 17, 858

 Response: We have added commented and include on the reference (#85); line 267, 268.

Line 152 the numeration of chapter should be changes, In my opinion “Non-phenol red beetroot components with chemopreventive activity” should be another, separate chapter 2.4, not “Polyphenols” part

Response: Corrected

More important information on the anticancer mechanism of individual groups of phytochemicals should be collected in the tables with appropriate reference.

Response: We have included a Table

References:

Line 293: title of journal in reference no. 3 should be corrected

Response: Corrected

Line 398: title of journal should be italic

Response: Corrected

Reviewer 2 Report

The manuscript contains useful information. However, it needs thorough rearrangement with paragraphs content defined or described briefly in the first sentence. 

Also caretenoids should be a separate section rather than being subsection of phenolics. 

 Furthermore, conclusions should be precise and mostly devoid of citations unless until necessary.

Authors should also add a figure sequencing the different cellular events mainly defining or affecting cancer process and symptoms and diagnostics.

Also, author can briefly mention the summarized effects of the components in a Table, which makes it easier to see and increases the appeal of the manuscript. The different effects of same component can be mentioned in bullets.

Author Response

The manuscript contains useful information. However, it needs thorough rearrangement with paragraphs content defined or described briefly in the first sentence. 

Also caretenoids should be a separate section rather than being subsection of phenolics. 

 Furthermore, conclusions should be precise and mostly devoid of citations unless until necessary.

Response: Carotenoids now have a separate section.  We feel the references in the Conclusions Section are necessary for understanding by the reader.

Authors should also add a figure sequencing the different cellular events mainly defining or affecting cancer process and symptoms and diagnostics.

Response: Figure has been added

Also, author can briefly mention the summarized effects of the components in a Table, which makes it easier to see and increases the appeal of the manuscript. The different effects of same component can be mentioned in bullets.

Response: Table has been added

Round  2

Reviewer 2 Report

The authors have done the minor revisions; however, the manuscript needs further modifications.

Table 1 should have columns with different components, their beneficial effects, and citations. This table is a summary of all components, which could also have been summarized in a sentence. Authors are advised to modify.

Change the title conclusion to “conclusions and future research”.

Authors should also change the position of figure 1, in the middle of the text about Betanins.

Author Response

The authors have done the minor revisions; however, the manuscript needs further modifications.

Table 1 should have columns with different components, their beneficial effects, and citations. This table is a summary of all components, which could also have been summarized in a sentence. Authors are advised to modify.

Response:  We have developed the table with the information and formant suggested.

Change the title conclusion to “conclusions and future research”.

Response:  We have changed the title of the Conclusions section to Conclusions and Future Research.

Authors should also change the position of figure 1, in the middle of the text about Betanins.

Response:  We have changed the position of Figure 1.